# Fetus Exposure to Drugs and Chemicals: A Holistic Overview on the Assessment of Their Transport and Metabolism across the Human Placental Barrier

**DOI:** 10.3390/diseases12060114

**Published:** 2024-06-01

**Authors:** Ioly Kotta-Loizou, Agathi Pritsa, Georgios Antasouras, Spyridon N. Vasilopoulos, Gavriela Voulgaridou, Sousana K. Papadopoulou, Robert H. A. Coutts, Eleftherios Lechouritis, Constantinos Giaginis

**Affiliations:** 1Department of Life Sciences, Faculty of Natural Sciences, Imperial College London, London SW7 2AZ, UK; i.kotta-loizou2@herts.ac.uk; 2Department of Nutritional Sciences and Dietetics, School of Health Sciences, International Hellenic University, 57400 Thessaloniki, Greece; gabivoulg@gmail.com (G.V.); sousana@the.ihu.gr (S.K.P.); 3Department of Food Science and Nutrition, School of the Environment, University of the Aegean, 81400 Lemnos, Greece; g.antasouras@gmail.com (G.A.); fns10059@fna.aegean.gr (E.L.); cgiaginis@aegean.gr (C.G.); 4DNA Damage Laboratory, Physics Department, School of Applied Mathematical and Physical Sciences, National Technical University of Athens (NTUA), Zografou Campus, 15780 Athens, Greece; svasilopoulos@acg.edu; 5Department of Clinical, Pharmaceutical and Biological Sciences, School of Life and Medical Sciences, University of Hertfordshire, Hatfield AL10 9AB, UK; r.coutts@herts.ac.uk

**Keywords:** placental barrier, passive diffusion, active transport, metabolism, fetotoxicity, ex vivo perfusion, in vitro studies, in vivo animal studies, placental analysis, placental metabolomics

## Abstract

Background: The placenta exerts a crucial role in fetus growth and development during gestation, protecting the fetus from maternal drugs and chemical exposure. However, diverse drugs and chemicals (xenobiotics) can penetrate the maternal placental barrier, leading to deleterious, adverse effects concerning fetus health. Moreover, placental enzymes can metabolize drugs and chemicals into more toxic compounds for the fetus. Thus, evaluating the molecular mechanisms through which drugs and chemicals transfer and undergo metabolism across the placental barrier is of vital importance. In this aspect, this comprehensive literature review aims to provide a holistic approach by critically summarizing and scrutinizing the potential molecular processes and mechanisms governing drugs and chemical transfer and metabolism across the placental barrier, which may lead to fetotoxicity effects, as well as analyzing the currently available experimental methodologies used to assess xenobiotics placental transfer and metabolism. Methods: A comprehensive and in-depth literature review was conducted in the most accurate scientific databases such as PubMed, Scopus, and Web of Science by using relevant and effective keywords related to xenobiotic placental transfer and metabolism, retrieving 8830 published articles until 5 February 2024. After applying several strict exclusion and inclusion criteria, a final number of 148 relevant published articles were included. Results: During pregnancy, several drugs and chemicals can be transferred from the mother to the fetus across the placental barrier by either passive diffusion or through placental transporters, resulting in fetus exposure and potential fetotoxicity effects. Some drugs and chemicals also appear to be metabolized across the placental barrier, leading to more toxic products for both the mother and the fetus. At present, there is increasing research development of diverse experimental methodologies to determine the potential molecular processes and mechanisms of drug and chemical placental transfer and metabolism. All the currently available methodologies have specific strengths and limitations, highlighting the strong demand to utilize an efficient combination of them to obtain reliable evidence concerning drug and chemical transfer and metabolism across the placental barrier. To derive the most consistent and safe evidence, in vitro studies, ex vivo perfusion methods, and in vivo animal and human studies can be applied together with the final aim to minimize potential fetotoxicity effects. Conclusions: Research is being increasingly carried out to obtain an accurate and safe evaluation of drug and chemical transport and metabolism across the placental barrier, applying a combination of advanced techniques to avoid potential fetotoxic effects. The improvement of the currently available techniques and the development of novel experimental protocols and methodologies are of major importance to protect both the mother and the fetus from xenobiotic exposure, as well as to minimize potential fetotoxicity effects.

## 1. Introduction

The placenta exerts a crucial impact in supplying nutrients and oxygen to the fetus through the umbilical cord, which acts as a membrane barrier between the maternal and fetal bloodstream [1]. This organ possesses important roles during gestation, such as fetal nourishment with glucose, amino acids, fatty acids, minerals, vitamins, and oxygen, as well as support and protection, gas exchange, and the production of several hormones and other mediators [1,2]. Furthermore, several steroids and polypeptide hormones and placental galactogen are produced in the placenta, which can effectively contribute to the survival and growth of the fetus [3]. These placental hormones are vital for gestation establishment and maintenance, regulating decidualization, placental development, angiogenesis, endometrial receptivity, embryo implantation, immunotolerance, and fetal development [4]. Any disruption of the aforementioned processes can cause placental dysfunction, which in turn could lead to placental abnormalities, implying the dysregulation of diverse molecular pathways, thus inducing a variety of disorders during pregnancy, including failed abortion, spontaneous abortion, intrauterine growth restriction (IUGR), oxidative stress, excessive accumulation of dysfunctional adipose tissue related to insulin resistance, diabetes, and other metabolic disorders, gestational hypertension, and pre-eclampsia [5,6,7,8]. Characteristically, abnormal placentation is not only related to direct deleterious consequences for the outcome of gestation, but also predisposes the offspring to metabolic, cardiovascular, and neuropsychiatric disorders and certain types of tumor malignancies during adult life, which may be ascribed to diverse structural and epigenetic alterations in organ systems [8,9].

During recent decades, there has been increased research refuting the theory that the uterus provides a protected environment for the fetus, which is mainly attributed to the placental barrier [10,11]. The above process requires enhanced caution worldwide, as pregnant women and their developing fetuses are at risk of exposure to diverse drugs and chemicals like environmental pollutants and pesticides [12]. In this aspect, developmental toxicology data are systematically utilized for the prioritization and screening of drugs and chemicals, for the assessment and labeling of drugs, and for describing hazards and risks of exposure to industrial and environmental chemicals [12,13]. Notably, during the initial weeks of gestation, women do not necessarily know they are pregnant and therefore do not worry about their gestational state; however, this may result in unintended exposure to diverse toxic xenobiotic compounds. Notably, prescription medication is usually inevitable during pregnancy. However, the list of drugs that can be taken during pregnancy is severely limited due to the sensitivity of the placental barrier.

In addition, pregnant women suffering from some disease, such as a metabolic disorder and/or epilepsy, are frequently exposed to pharmaceuticals throughout gestation [13,14]. There are also additional risk factors of exposure, including certain lifestyle factors, such as smoking, the use of daily care products, alcohol intake, and recreational drugs, together with environmental pollutants, pesticides, heavy metals, and other synthetic compounds [10,11]. Moreover, fetal treatment through maternal drug administration is even more frequently employed, such as the retroviral treatment received by pregnant women with HIV [12,13]. Notably, nanomedicine constitutes an exceptional opportunity to target and support drug delivery to the reproductive system and other relevant organs in both the mother and fetus, and it can effectively improve xenobiotics’ safety profile and minimize side effects [15].

There is currently a great deal of evidence reinforcing the fact that xenobiotic compounds can penetrate the placental barrier by passive diffusion and/or active transport mechanisms. These compounds can be metabolized in the presence of placental xenobiotic-metabolizing enzymes and produce metabolites that are potentially toxic to the fetus. Thus, a thorough discussion of pharmacological and chemical exposure to the fetus, as well as transportation and metabolism through the human placenta, is highly recommended. In this context, the present comprehensive review aims to critically summarize and scrutinize the potential molecular mechanisms governing maternal xenobiotics’ exposure during pregnancy, which may subsequently facilitate their transport and metabolism across the placental barrier, leading to deleterious effects on the fetus. The elucidation and the understanding of the above molecular processes and mechanisms could contribute to the promotion of both the mother’s and fetus’s health, avoiding potential fetotoxicity effects.

## 2. Methods

A comprehensive literature review was conducted using the most accurate scientific databases, e.g., PubMed, Scopus, and Web of Science, applying effective, characteristic, and relevant keywords such as “drugs”, “chemicals”, “placenta”, “fetotoxicity”, “pregnancy”, “placental transfer”, “placental metabolism”, “gestation”, “pollutants”, “placental barrier”, “in vitro studies”, “in vivo animal studies”, “ex vivo studies”, etc. Inclusion criteria were any studies written in the English language, clinical human studies, in vitro and in vivo animal studies, ex vivo perfusion studies, and computer-aided studies. Gray literature, commentaries, editorials, letters to the editor, abstracts in conference proceedings, and articles in non-peer-reviewed journals were excluded from the final analysis. Papers were also excluded if they did not fit into the conceptual framework of this study. The search was supplemented by scanning the reference lists of relevant studies and manually searching key journals, commentaries, editorials, and abstracts in conference proceedings. No time limitation was applied for the final selection of the studies included in the present review. The retrieved surveys were additionally comprehensively checked for related studies quoted in their text. All the authors acted as reviewers. To enhance reliability among reviewers, all reviewers screened all the recovered studies, discussed their findings, and edited the screening and data extraction manual before starting screening for this review. All the reviewers working in pairs sequentially assessed the titles, abstracts, and then the full text of all studies detected by our searches for potentially relevant publications. We resolved disagreements on study selection and data extraction by consensus and discussion with all the authors/reviewers if required. A first search in the three scientific databases revealed 8830 published articles until 5 February 2024. After applying all the above exclusion criteria in conjunction with the articles retrieved from the full text of the initial selected articles, 148 published articles were included in the present review study. A flow chart diagram for study enrolment employing the PRISMA guidelines is depicted in Figure 1.

## 3. Results

### 3.1. Placenta Anatomy and Function Characteristics Related to Placental Transfer and Metabolism Mechanisms

The placenta is a tentative and extremely complex organ that directly connects the mother to the fetus. There are two main cell lines in mammals that may be involved in the formation of the placenta. More to the point, primary cell differentiation occurs during the formation of the trophiderm [16]. In fact, the trophoblast cell lineage can establish the epithelial placental parts, while the extraembryonic mesoderm participates in the development of the stromal cells and blood vessels [17]. In the innermost placenta layer, there are also formations of villous and tree-like branches (chorionic villi), offering a surface for nutrient and gas exchange [16,17]. In addition, villous “trees” constitute the basic structure of the placenta. Based on the developmental stage, the villous structure, vessel branches, histologic features, and vessel-cell-type components are gradually developed [16,17]. The villi constitute the primary xenobiotics’ transport site within the mature placenta, being formatted in the lacunar period (8–13 days post-conception) [16,17,18]. The human uterus is composed of the myometrium, a smooth muscle layer; the endometrium, an inner mucosal lining surrounding the mesenchymal stroma; immune cells; and an extensive vascular network [16,17,18]. The endometrium is further divided into the deep stratum basalis, adjacent to the myometrium, and the superficial stratum functionalis [18,19]. The weight of the placenta is about 20 g at the 10th week and 150–170 g at the 20th week of gestation. A mature placenta weighs about 500–600 g and consists of 15–28 cotyledons [18,19]. In Figure 2, a representative illustration of placenta morphology is depicted.

Placental angiogenesis is crucial to maintain sufficient blood flow during gestation, and any alteration in this process may result in an adverse pregnancy [20]. The mammalian placenta releases multiple angiogenesis molecules, such as vascular endothelial growth factor (VEGF) [18,19,20]. Increasing evidence has suggested that suboptimal maternal nutrition can alter placental development. Peroxisome proliferator-activated receptors (PPARs), which can be activated by ligands including long-chain polyunsaturated fatty acids, affect placental angiogenesis directly through angiogenic factors or indirectly by modulating trophoblast differentiation [20,21]. Maternal-to-fetal transport of substances within the syncytiotrophoblast occurs across the brush border membrane and then across the basal membrane and vice versa from the fetal to maternal bloodstream [18,20]. Uterine vascular remodeling is intrinsic to the cycling of the endometrium and the early pregnant endometrium [20,21]. Maternal regulatory factors such as ovarian hormones, VEGF, angiopoietins, Notch, and uterine natural killer cells significantly mediate these vascular changes [22,23]. The placental barrier or maternal–fetal diffusion distance diminishes during gestation from 50 mm at the end of the 2nd month to 5 mm by the 37th week of gestation [18,19,23]. Furthermore, the syncytiotrophoblast layer thins during gestation, while the number of microvilli increases in the third trimester, promoting material exchange between mother and fetus [23,24].

### 3.2. Evaluating Placental Transfer as a Method for Estimating Fetal Toxicity

Assessing xenobiotic transport within the human placental barrier is of fundamental importance to ensure drug and chemical safety during gestation. However, placental transfer does not necessarily result in fetotoxic effects but can also play a protective role against diverse toxic xenobiotics. Notably, the transfer through the placenta can serve as a valuable factor for describing distribution within the human placenta. Understanding placental transfer, alongside other influencing factors such as absorption and metabolism, is often advised to precisely assess the potential risks to the developing fetus [25]. In Table 1, the most important advantages and disadvantages of placental transfer assessment methods are presented.

#### 3.2.1. In Vitro Studies

The human trophoblast barrier can be applied to explore the exchange of xenobiotics between the maternal and fetal compartments, as well as by exploring intracellular metabolism, paracellular contributions, and regulatory mechanisms, which can affect the vectorial transport of molecules. Several in vitro models such as primary trophoblast cells, placental immortal cell lines, and placental explants have frequently been used [26,27,28]. In vitro techniques constitute an effective alternate approach simulating in vivo animal models, as the latter approach is reasonably more costly and raises serious ethical issues [18,19,20]. No differentiated human cytotrophoblasts are initially developed and they spontaneously syncytialize and become an appropriate in vitro model to explore uptake, efflux, metabolism, and hormone release [29,30]. The above non-proliferative, multinucleated cells are characterized by elevated intercellular spaces, being developed on semi-permeable membranes. To overcome this problem, a study merged syncytiotrophoblasts’ cell layers on semi-permeable supports. These cell layers of syncytiotrophoblasts can act as a sufficient barrier concerning both low- and high-molecular-weight xenobiotic compounds [30,31].

Immortalized trophoblast cell lines constitute another alternative and appropriate in vitro experimental approach. This type of cell line is derived from both normal and malignant tissue or embryonic carcinomas, reinforcing the need for trophoblast differentiation [32,33]. In fact, cell lines derived from human choriocarcinoma can effectively stimulate the endocrine activities of human trophoblasts [33,34]. In addition, it is considered quite easy to obtain a sufficient simulation of the placental barrier from placental choriocarcinoma cell lines, such as BeWo, Jar, and JEG-3, by permitting cells to construct both monolayers and multiple layers [35,36,37]. Indicatively, an in vitro study utilizing the BeWo cell line showed that synthetic cannabinoids can lead to apoptosis and mitochondrial dysfunction in placental cells, thus leading to disturbances in fetus health [38,39]. Typically, the BeWo cell line has successfully been utilized in exploring drug transportation. Moreover, BeWo cell line transport rates are adequately associated with those in the ex vivo perfusion model [38,39].

Human trophoblastic tissue is commonly employed to investigate amino acid uptake [40]. Similarly, utilizing placental tissue cultured as tissue explants offers numerous advantages for investigating placental transport, endocrine function, and cellular proliferation and differentiation. [41]. Moreover, a crucial advantage of placental tissue cultures concerns the intact microarchitecture, while parallelly ensuring the intercell and paracrine communications [41]. Nonetheless, trophoblasts in tissue explants exhibit low viability, which cannot be guaranteed by laboratory culture conditions, whereas, concurrently, cell viability in perfused placenta appears to be superior [42]. Lastly, cell cultures obtained by amniocentesis and chorionic villus sampling appear to be quite suitable to investigate prenatal cell toxicity and the risk of chromosomal abnormalities [43,44].

#### 3.2.2. Ex Vivo Perfusion Techniques

The ex vivo human placental perfusion approach is widely regarded as the most dependable approach for assessing placental transfer and metabolism. Initially presented by Schneider et al. [45], this method has since been refined in subsequent studies [46].

Ex vivo perfusion of the human full-term placenta provides a means to directly investigate placental transfer, eliminating the need for extrapolation from animal models and thereby sidestepping ethical concerns for both the mother and her fetus. It represents a more accurate and informative technique compared to in vitro methods [47]. Crucially, it preserves the placental structure entirely, enabling comprehensive insights into various transporters associated with active transport and tissue binding, albeit being somewhat time-consuming [47,48]. Additionally, by perfusing placentas from mothers who smoke, misuse drugs, or have a diagnosed disease, researchers can investigate the impact of these factors on fetal exposure to genotoxic agents [47,48]. Indicatively, the commonly used herbicide diuron is metabolized to a toxic metabolite more readily available in placentas from regular smokers compared to non-smokers [46,47,48].

Xenobiotic transport is commonly assessed by comparing xenobiotics to the reference compound antipyrine, allowing a direct comparison of experimental data across different laboratories. Antipyrine constitutes a small lipophilic molecule that can solely pass through the placenta by passive diffusion, as it does not bind to proteins [46,47,48]. Additionally, antipyrine acts as an internal reference point to reduce the risk of confusion between the fetal and maternal bloodstream, standardizing variations between different placental samples due to factors like blood flow [46,47,48]. The quantification of placental transport is presented either as a Transfer Index (TI) or a clearance index (CI), both determined using antipyrine as a reference compound: TI = transfer of drug (%)/transfer of antipyrine (%) and Cl = clearance of drug/clearance of antipyrine.

Beyond the inter-individual dissimilarities among humans, it remains unclear how many placentas should be perfused to effectively overcome these variations, which constitutes another important issue [49]. In addition, the cells of perfused placentas remain viable and feasible for up to 12 h [20,21]. Thus, while this model is applicable for diverse investigations, it is not suitable for the longer perfusion periods that are required to study the transmission of infectious diseases [20,21]. Additionally, it typically utilizes full-term placentas, which means that it does not accurately represent the first-trimester placenta [44,48], leading to differences in transporters and metabolic enzymes compared to full-term placentas. Another limitation is the reliance on fresh placentas for ex vivo human placental perfusions, necessitating a consistent supply of placentas [46,47,48]. Moreover, it is a time-consuming procedure with a relatively low success rate, posing challenges for large-scale studies at a population level. Generally, the datasets published using this procedure remain small; however, strict control of fetal blood pressure, flow, and glucose and maternal pH has an almost two-fold success rate [43,44]. Moreover, the ex vivo perfusion system is metabolically stable in contrast to the gestational state. Based on the assessment of multiple quality control measurements, a closed human ex vivo placenta perfusion model was validated with a more than doubled success rate (38%) compared to that reported in the literature (15%) [20,21,22].

#### 3.2.3. Animal Studies

Many in vivo studies have been carried out using pregnant animal models to explore the potential for birth defects. In this context, a promising newcomer among rodents considers the spiny mouse, which has a longer gestational duration than the mouse with organogenesis completed at birth. The guinea pig is also recommended because there is a plethora of information concerning its gestation and placentation in the literature. Several smaller primates have also been considered [32,33,34]. More to the point, the marmoset placenta has some features that closely resemble human placentation, such as the interhaemal barrier [32,33,34]. Several similarities were drawn between mice and humans regarding placental cell types and genes controlling placental development. Nevertheless, substantial differences, including a different kind of implantation, a prominent yolk sac, and lower levels of placental hormones, have been reported in mice [49]. Guinea pigs have also been identified as a representative alternative rodent model together with rats [48,49]. Thus, pregnancy research could greatly benefit from increased use of alternative models such as the spiny mouse and common marmoset [48,49,50]. On the other hand, sheep appear to be of inadequate value for placental research [48,49,50]. However, a serious concern is raised concerning the relevance of extrapolating experimental data obtained in animal placental studies to human situations [18,48].

The limitations mentioned above primarily stem from the distinct structural and functional differences between the human placenta and those of other mammals [35,36,37,38]. The human placental barrier comprises a single layer of multinucleated cells known as syncytiotrophoblasts. During human placental development, embryonic tissues are situated nearer to the maternal surface of the lamina, which is coated with a layer of syncytiotrophoblasts interspersed with precursor cells referred to as cytotrophoblasts [51,52]. Therefore, although there is growing evidence towards utilizing whole human placental tissue extracts and cells, animal studies continue to be crucial for evaluating the fetotoxic impact of xenobiotic compounds [48,50].

The above models provide quite sufficient data about placental transfer and the potential toxic effects of xenobiotics on the fetus [53,54,55]. In this aspect, novel challenges in designing therapeutics for treatment potential complications during gestation, such as nanomedicine, involve selectively delivering payloads to the placenta by simultaneously protecting the fetus from potential toxic side effects. In this context, murine models have been used to explore the fetal bioaccumulation and toxicity of maternally administered nanomaterials in an age- and dose-dependent manner [53,54,55]. Moreover, in pregnancy, nanotherapeutic delivery offers novel and effective options to stably deliver silencing RNA and microRNA inhibitors to the placenta to regulate gene expression, reinforcing novel genetic treatments for pre-eclampsia and fetus growth restriction. Thus, the development of targeted therapies, utilizing nanoparticles in a reproductive setting, may ultimately permit safe and focused treatments for conditions affecting the health and reproductive capacity of women as well as for the management of gestation and its serious complications [53,54,55]. In recent years, nanomaterials are widely used in medicine for the diagnosis and treatment of a broad range of diseases, including cancer and other disorders. Indicatively, iron oxide nanoparticles have attracted much attention because of their potential applications, which are attributed to their small size and their potential to cross the placental barrier [53,54,55]. However, the above studies have been focused on both positively and negatively charged iron oxide nanoparticles, as they have the ability to cross the placenta and accumulate in the fetus. Thus, high bioaccumulation and toxicity were observed with a positively charged surface coating, highlighting the strong demand for further research in this promising research field.

#### 3.2.4. Human Studies

Because of ethical considerations, studies assessing the risks associated with maternal drug and chemical exposure in humans are strictly limited. The assessment of the feto-maternal concentration ratio (F/M) is the most frequently used method for monitoring drugs and treating pregnant women with serious pathological conditions, such as metabolic and neurodevelopmental disorders, who are undergoing medication treatment or recommended fetal therapy via maternal drug monitoring. This approach represents a straightforward and ethically acceptable technique. Nonetheless, there is currently no sufficient information concerning drug or chemical distribution within tissues based on such data, highlighting the need for further research on this issue [18,19,20,21,22].

Several endogenous and/or exogenous factors can significantly affect placental function, leading to considerable inter-individual variations. As gestation advances, the placenta becomes increasingly vulnerable to xenobiotic agents because of decreases in placental thickness and a reduction in the number of cell layers. Moreover, the transport of xenobiotics can be affected by diverse maternal and/or fetal pathologies, leading to impaired placental function. In cases of pre-eclampsia, fetal cytotrophoblast cells typically exhibit limited invasion into the maternal artery system’s decidual portions. However, they cannot endure the adhesion receptor switching characteristics of normal gestation. The above findings may result in lowered placental perfusion [56,57].

#### 3.2.5. Analytical Techniques for Evaluating Xenobiotic Maternal and Fetal Exposure

Analytical methodologies and techniques have a crucial role in identifying and quantifying xenobiotic compounds to which mothers and fetuses are exposed. Since the 1980s, gas chromatography–mass spectrometry (GC-MS) has been considered as the gold standard in analytical toxicology with selected ion monitoring (SIM) for immunoassay confirmation, targeted screening, and quantification [58]. In the 1990s, liquid chromatography–mass spectrometry (LC-MS) with electrospray ionization (ESI), atmospheric pressure chemical ionization (APCI), or atmospheric pressure photoionization (APPI) revolutionized bioanalysis in analytical toxicology [59]. LC coupled to tandem MS (LC-MS/MS) with selected reaction monitoring (SRM) for targeted (multi-analyte) screening and quantification or with data-dependent or data-independent product ion spectrum formation for comprehensive screening has been considered as a new gold standard [58,59]. The next trend began in recent years with the coupling of high-resolution mass spectrometry (HRMS) mostly with GC or LC for the analysis of small and large molecules in analytical toxicology [60]. HRMS was developed in the 1960s with double-focusing mass spectrometers, but today, time-of-flight (TOF) or Orbitrap (OT) mass analyzers are common, mostly as hybrids with triple quadrupoles (QTOF, QOT) or ion traps in front, allowing fragmentation to reproducible MS/MS spectra [61].

The above analytical techniques have increasingly been used to determine the amounts of xenobiotics in the biological fluids of pregnant women and their fetuses. These techniques have become even more sensitive and reliable with lower detection limits in the last few years. Characteristically, ultra-performance liquid chromatography tandem mass spectrometry (UPLC-MS/MS) was applied to detect 20 perfluoroalkyl substances, which can cross the placental barrier in 424 mother–fetus pairs [62]. In utero exposure levels and transplacental transfer characteristics of the ubiquitous environmental pollutant polybrominated diphenyl ether in paired human samples (30 placenta and 30 cord blood samples) were quantified and analyzed using gas chromatography/mass spectrometry (GC/MS) [63]. UPLC-MS/MS was also used to explore the placental transfer of Letermovir and Maribavir in the ex vivo method of the human perfused cotyledon [64]. GC coupled with triple quadrupole mass spectrometry (QqQ-MS/MS) has further been utilized to assess prenatal exposure to diverse environmental organic pollutants, such as polycyclic aromatic hydrocarbons, organophosphorus pesticides, polybrominated diphenyl ethers, and others, using non-invasive biological samples (meconium and placenta) [65]. LC-MS/MS was also effectively used to obtain more knowledge about the placental transfer of uremic solutes across the human placenta [66]. GC coupled to tandem MS was applied to detect polybrominated diphenyl ethers and polychlorinated biphenyls in the placentas of mothers with gestational diabetes mellitus [67]. In addition, UPLC-MS/MS was utilized to estimate maternal-to-fetal transfer clearance of ciprofloxacin at a therapeutic concentration and to determine fetal exposure to maternally administered ciprofloxacin [68]. LC-MS was used to evaluate darunavir and cobicistat pharmacokinetics during pregnancy compared with the postpartum period and in infant washout samples after delivery [69]. The UHPLC-MS/MS method was applied for the simultaneous quantification of 15 selected bisphenols in perfusion media of an ex vivo human placental perfusion model [70].

More recently, validated high-performance liquid chromatography–ultraviolet (HPLC-UV) and LV-MS assays were used to assess atazanavir and cobicistat pharmacokinetics during pregnancy compared with the postpartum period and in infant washout samples [71]. Another analytical technique, inductively coupled plasma mass spectrometry (ICP-MS), was applied to directly assess placental iron transport and bypass maternal intestinal iron absorption [72,73]. In addition, UHPLC-MS/MS coupled with multisite microdialysis was used to monitor the levels of remdesivir and the nucleoside analog GS-441524 in the maternal blood, fetus, placenta, and amniotic fluid of pregnant Sprague Dawley rats [73]. HPLC-MS/MS was also utilized to determine the levels of legacy per- and polyfluoroalkyl substances in 50 paired samples of maternal and cord serum of pregnant women at delivery [74]. Moreover, UHPLC-MS/MS combined with a microdialysis system was developed to monitor codeine, morphine, and morphine-3-glucuronide in multiple sites of maternal blood, placenta, fetus, and amniotic fluid after codeine administration [75]. Notably, nano-LC-MS/MS was applied to determine nutrient transport expression in human fetal membranes and fetal membrane cells and compared expression with placental tissues and BeWo cells [76]. Mishra and Kumar used LC-MS to evaluate placental long-chain polyunsaturated fatty acid transport and metabolism in a rat model of lean gestational diabetes mellitus [77]. UHPLC-MS/MS has recently been applied to assess whether molnupiravir and the nucleoside analog β-D-N4-hydroxycytidine could cross the blood–placenta barrier into the fetus [78]. ICP-MS was also used to evaluate the placenta transfer of the trace metal cobalt (Co) from mother to fetus in 246 mother–infant pairs from a case–control study on birth defects [79]. Additionally, UPLC-MS/MS was applied to determine the drug concentration of etomidate, remifentanil, and rocuronium bromide for general anesthesia in fetuses as well as the placental transport rate between full-term and pre-term delivery, twins, and singletons [80]. ICP-MS was also utilized to investigate the effect of the placental barrier on the transfer of trace elements from mother to fetus and its relationship with hypertensive disorders complicating pregnancy and gestational diabetes mellitus [81].

### 3.3. Placental Transport Mechanisms

Xenobiotics are either transported freely in plasma or bound to carrier proteins through the maternal bloodstream. During transplacental transport, diverse endogenous substances as well as xenobiotics can enter the co-cytotrophoblast layer. This can occur through passive diffusion across the trophoblast cytoplasm or via a complex network of active transporters. Additionally, plasma protein binding (e.g., with albumin, g-glycoprotein, etc.) can decrease placental transport, as only the unbound xenobiotic passes through the placental barrier, which can enhance placental transport by passive diffusion mechanisms. The role of efflux and influx transporters, which are located on the brush border and/or basal membrane side of the syncytiotrophoblast, is additionally of fundamental importance [57,82].

#### 3.3.1. Active Transport

The placenta contains a variety of transporters that can either facilitate or impede the passage of xenobiotics from the mother to the fetus (see Figure 3). These transporters can also recognize various endogenous substances, such as essential nutrients or metabolic by-products. The directional transport may be attributed to the different expression of diverse transporters in the maternal-facing brush border membrane and the fetal-facing basal membrane [20,83,84]. These transporters can exert a crucial role in supplying essential nutrients to the fetus and removing metabolic by-products. Conversely, they can interact with xenobiotics that share a similar chemical structure with endogenous substrates, thus facilitating their transfer from mother to fetus or safeguarding the fetus [83,84]. This phenomenon depends on the specific location (apical or basolateral) of each placental transporter. Additionally, a significant clinical concern is that many xenobiotics can influence the gene expression of certain transporters, potentially altering their physiological functions and the mechanisms of the entry of drugs or chemicals into the placenta [20,83,84]. For example, polystyrene nanoparticles can actively be transported from the fetus to the mother [83,84].

The ATP-binding cassette-containing (ABC) family of proteins includes active drug efflux transporters like P-glycoprotein (P-gp), multidrug resistance-associated proteins (MRPs), and breast cancer-resistant protein (BCRP), which can exert a significant impact on placental transfer [85,86]. These transporters typically function in conjunction with ATP hydrolysis. The most well-known active transporter is P-gp, which is located in the syncytiotrophoblast (apical membrane—maternal side), acting as a pump transport molecule from inside the cell to the extracellular space [85,86] and also functioning as a barrier by protecting the fetus when exposed to xenobiotics. [83,84]. P-gp can transport drugs and chemicals with a variety of molecular structures, but all are hydrophobic and amphipathic, such as anticancer, immunosuppressive, and cardioprotective drugs, as well as steroids and lipid-lowering agents [48,49]. Pharmacological blocking of P-gp, such as with calcium channel blockers, used to treat gestational hypertension, can enhance this protection [85,86,87]. Conversely, inhibiting P-gp can increase drug levels in the fetus, as seen in mothers with HIV receiving antiretroviral therapy or in cases of a sick fetus [87,88].

It has been shown that during pregnancy, mRNAs encoding MRP1, 2, 3, and 5 are detected in the human placenta [89]. Both MRP1 and MRP3 transporters are situated on the basolateral membrane of the syncytiotrophoblast (fetal side) [90]. On the other hand, MRP2 is located in the apical membrane of the syncytiotrophoblast (maternal side), while MRP5 is in the basal membrane of syncytiotrophoblasts (fetal vessels) but also in the apical membrane. Both MRP1 and MRP3 function as organic anion transporters and prevent the entry of glutathione/glucuronide metabolites into the fetus since they assist in their excretion [89,90]. The role of MRP1, MRP2, and MRP3 is still under investigation and may also contribute to the efflux of polar conjugates [56,57,82,83,84] in addition to glucuronide transport. As for MRP5, it seems to be involved in the transport (excretion) of cyclic nucleotides, especially of cGMP, thus influencing the differentiation of cytotrophoblasts [89,90].

BCRP was initially found to provide resistance against anthracycline antitumor drugs and mitoxantrone [90,91]. Many drugs commonly administered during pregnancy, such as nitrofurantoin, cimetidine, and glyburide, are substrates of BCRP [91,92]. BCRP is situated on the apical membrane of the syncytiotrophoblast (maternal side), and its expression levels in the placenta vary throughout gestation. Pregnancy-related steroid hormones, growth factors, and cytokines have also been considered as potential regulators of BCRP expression in the human placenta [89,90,91,92]. BCRPs’ structure, substrate selectivity, and localization in the placenta suggest that, like P-gp, they could exert a protective role in removing cytotoxic drugs or chemicals from the fetus [92,93].

Transporters such as organic anion transporters (OATs), organic cation transporters (OCTs), and neuronal monoamine transporters (NMTs) found in the syncytiotrophoblast membrane can provide protection to the fetus [93,94,95]. In fact, OATs transfer organic anions, which are localized into and out of cells, and they have very broad substrate specificity, including steroid sulfates derived from cigarette smoking [93,94,95]. Moreover, several members of the OATPs, such as OATP1A2, OATP1B1, OATP-2B1, OATP-3A1, and OATP-4A1, which participate in the Na^+^-independent uptake of bile acids in the liver, have also been detected in the placenta. In addition, OATP2B1 was further shown to contribute to the transepithelial transport of steroid sulfates in the human placenta [92,93,94,95]. Some data suggest that OATPs could play a role in the transport of drugs such as anticancer drugs (e.g., methotrexate), non-steroidal anti-inflammatory drugs, and antibiotics [92,93,94,95]. Three OCTs are expressed in the human placenta, namely OCTN1, OCTN2, and OCTN3. OCTN1 facilitates the transport of carnitine, an endogenous metabolite produced during the oxidation of fatty acids, as well as various drugs including procaineamide and ofloxacin [92,93,94,95]. OCTN2 operates as a transporter bound to sodium (Na^+^), facilitating the transfer of endogenous carnitine from the mother to the embryo, as well as several exogenous substances, including clonidine and verapamil [93,94,95]. The primary function of OCTN3 is the removal of catecholamines from the fetal circulatory system [93,94,95].

Serotonin, dopamine, norepinephrine, and epinephrine constitute the most important physiological substrates of the NMTs in the brush border membrane of the placenta [93,94,95,96]. Remarkably, several antidepressants, along with cocaine, act as inhibitors of NMTs without being transportable substrates themselves. Conversely, amphetamines are substrates that can be transported, inhibiting NMTs’ capacity to transfer endogenous substrates by acting as competitive inhibitors [93,94,95,96]. Lastly, diverse amino acid transporters (AATs) appear to facilitate the transportation of xenobiotics to the fetal compartment. For instance, some AATs appear to contribute to the transport of certain pharmaceuticals that have a similar chemical structure to amino acids, such as gabapentin [95,96,97].

Both the equilibrative nucleoside transporters (ENTs) and the concentrative nucleoside transporters (CNTs), two families of nucleoside transporters, contribute to the Na^+^-independent uptake of purine and pyrimidine nucleosides. ENT1 and ENT2 are also present at the placental barrier [92,93,94,95,96]. Specifically, ENT1 is situated in the apical membrane of syncytiotrophoblasts, while ENT2 is located in the syncytiotrophoblast layer of the human full-term placenta. Both ENT1 and ENT2 are bidirectional transporters, allowing them to mediate both the influx and efflux of nucleosides [95,96,97]. The direction of transport is often determined by the transmembrane concentration gradients for the substrates. Additionally, both ENT1 and ENT2 are involved in the transport of antiviral nucleoside analog drugs, such as didanosine, zalcitabine, and zidovudine, as well as certain anticancer drugs like cytarabine and gemcitabine [95,96,97]. In Table 2, the most common transporters that can facilitate or impede the passage of drugs or chemicals from the mother to the fetus are reported.

#### 3.3.2. Passive Diffusion

Passive diffusion occurs bidirectionally, from mother to fetus and vice versa, as observed with the xenoestrogen bisphenol A [98]. This process typically depends on the specific characteristics and properties of each molecule such as lipophilicity, ionization, hydrogen bonding, and molecular weight. There are, of course, other factors that affect passive diffusion such as plasma protein binding, volume of distribution, and renal excretion, as shown in Figure 3. Data on the quantitative relationship between these factors and the placental transport of drugs are poor, while for 20 specific drugs of different structure, a parabolic or bilinear relationship between placental transport and lipophilicity has been suggested, expressed by the logarithm of octanol–water partition (logP) [99,100,101,102]. The molecular weight of drugs also seems to have an important role in their transport, as research has shown that drugs with a high molecular weight are associated with a low clearance index and, possibly, partial placental transport [99,100,101,102].

The influence of drugs’ and chemicals’ lipophilicity and/or molecular weight on placental transfer has been corroborated by several studies, although most of them involved small datasets [103,104]. More recently, the significant impact of physicochemical properties on the placental transfer process has been acknowledged through the use of larger datasets comprising structurally diverse drugs, employing quantitative structure–activity relationship (QSAR) methodology [103,104]. Notably, Hewitt et al. [105] initially established more “universal” models using a combined dataset of 78 structurally different compounds. The complexity of the placental transport mechanism was shown when applying a multiple regression equation, involving factors such as polarity (total polar surface area, TPSA) (negative effect), hydrophobicity (ethyl groups) of the molecule, and positively charged halogen atoms, which had a moderate statistical performance [105]. Another study using 94 different drugs also argued that specific physicochemical properties could be used to assess the risk of drug and chemical transport across the human placental barrier, namely that high molecular weight and high hydrogen bonding capacity reduce their transport across the placental barrier [106,107]. Thus, a partial least squares (PLS) model was developed that showed improved statistical performance [106,107]. Lipophilicity, molecular size, and number of chlorine atoms were included in this model [106,107].

The impact of physicochemical properties becomes more complex as physiological changes happen during the gestational period [108,109]. Protein binding alterations may be triggered during gestation as far as albumin and alpha1-acid glycoprotein concentrations in both maternal and cord blood are concerned [110,111]. Maternal serum albumin binding appears to decrease, which may lead to an increase in the free proportion of xenobiotics and, thereupon, to elevated placental transfer by passive diffusion [110,111]. The ratio of albumin concentrations between maternal and fetal serum varies throughout the gestational period, potentially influencing the equilibrium of xenobiotics between these circulations. It is worth noting that transfer of benzo(a)pyrene is higher when using physiological albumin concentrations [110,111]. Additionally, changes in pH during gestation should be taken into account when interpreting data regarding xenobiotic placental transfer, as fetal plasma and amniotic fluid tend to be slightly more acidic than maternal plasma [112,113]. Thus, it has been supported that the accumulation of basic drugs due to ion trapping may often result in fetal drug concentrations that exceed maternal plasma concentrations, leading to fetotoxic effects [112,113].

Fetus exposure to drugs and chemicals also depends on maternal pharmacokinetics, such as the volume of distribution, the rate of placental metabolism and excretion, and the hemodynamic changes induced in the mother during pregnancy [114,115]. For instance, although mycotoxin T-2 is actively transported, its metabolite HT-2 depends on passive diffusion [114,115]. During pregnancy, the volume of distribution, renal plasma flow, and glomerular filtration rate are increased, leading to lower drug concentration and elevated renal clearance of xenobiotics [116,117]. Placental aging is an additional factor that can affect the passive diffusion process. During the period of gestation, a reduction in the thickness and surface area of the placental barrier between maternal and fetal circulation due to the partial disappearance of the cytotrophoblast layer occurs [116,117]. These age-dependent alterations render xenobiotics’ transfer in full-term placenta not completely comparable to that of pre-term placenta [116,117].

### 3.4. Placental Metabolism

The metabolic processes that occur in the placenta (Figure 4) mainly concern steroid hormones. The enzymes sulfatase, 3β-hydroxysteroid dehydrogenase, aromatase, and 17β-hydroxysteroid dehydrogenase (type 1 and type 2) are some of the enzymes of interest for several researchers since they participate in the metabolism of steroid hormones, such as the biosynthesis of estrogen and progesterone [118,119]. It is well known that during pregnancy, the levels of estrogen and progesterone increase, and this has effects on the expression and activity of certain enzymes involved in metabolism [118,119]. Also noteworthy is the fact that the placenta enzymes can produce metabolites, which, through their interactions with DNA, may lead to DNA adducts [98] that could affect the health of the fetus [118,119].

Maternal xenobiotic metabolism can also be altered due to an elevation of endogenous hormone levels such as progesterone, which stimulates the microsomal oxidase system of the liver [120,121]. As a consequence of the above, an increase in the biotransformation (metabolism) of some drugs, such as phenytoin, can be observed [120,121]. On the other hand, some other drugs such as theophylline and caffeine in the presence of increased estradiol levels are associated with reduced output from the liver and suppression of the microsomal enzymes that metabolize the drugs. Many pesticides have also been reported to act as endocrine disruptors [120,121]. In Table 3, some common enzymes for the metabolism of specific drugs or chemicals are presented.

Smoking affects placental steroidogenesis, and in one study, the placentas of smoking mothers were shown to contain approximately half the amount of progesterone compared to the placentas of non-smoking mothers [122,123]. Cytochrome P450 monooxygenase CYP1A1 induction by maternal cigarette smoking or drug abuse has been convincingly demonstrated in several studies [122,123]. Notably, glutathione-S-transferase (GST) does not appear to be considerably increased due to maternal cigarette smoking [101,102,103,121,122], whereas UDP-glucuronosyltransferase (UGT) activity is stimulated by both cigarette smoking and alcohol intake [124,125]. These observations support evidence that there is an imbalance between phase I and II biotransformation reactions to chemical stress due to cigarette smoke exposure [124,125]. Also, the amounts of smoking-related DNA adducts in the placenta appear to inversely be related with the childbirth body weight [124,125]. However, there are currently no experimental data to link adverse fetus responses with increases in certain enzyme activities and DNA adducts.

Maternal medication therapy can also affect the enzyme activities of the placenta. For instance, maternal glucocorticoid therapy can suppress the activity of certain placental xenobiotic-metabolizing enzymes [126,127]. Indicatively, azidothymidine appeared to increase CYP1A1, CYP reductase, β-glucuronidase, and GST activities in full-term and first-trimester placentas [126,127]. Other studies have shown that phenobarbital has no effect on placental enzyme activity even if it can induce metabolic enzymes in the liver [128,129]. All the above findings lead to the conclusion that, as in the liver, so in the placenta, exogenous xenobiotic stimuli can affect only certain enzyme activities. Additionally, a discrepancy has been noted concerning the distribution of persistent organic pollutants (POPs) between mother and newborn, linked to metabolic capabilities and resulting in underestimation of fetus exposure to these pollutants [128,129]. Finally, placental adducts, induction of CYP1A1, and placental glutathione are factors that could be characterized as markers of environmental chemical stress since they are all associated with environmental pollution [128,129].

## 4. Discussion

There is considerable evidence that some drugs and chemicals can be transferred from the mother to the fetus through the placental barrier, resulting in embryotoxic effects [42,43]. Some specific characteristics of drugs and chemicals, such as low molecular weight, a relatively small number of hydrogen bonding sites, and increased lipophilicity, can increase the transport of these xenobiotics across the placenta barrier, which may occur through passive diffusion mechanisms [130,131]. Because some drugs and environmental pollutants show similarities to endogenous substances, they can also use placental active transporters to cross the maternal–fetal interface. A variety of xenobiotic-metabolizing enzymes contribute to the placental metabolism of several drugs and pollutants during pregnancy to produce metabolites that may have very different toxicity, but not necessarily greater than the parent compounds [11,112,132]. For this reason, there is growing interest in determining how safe it is for the fetus during pregnancy when such transport and metabolism of drugs and chemicals occurs across the placenta barrier.

Many studies have been carried out in the recent decades, in vivo and in vitro, as it is significant to know how the placenta functions both as a transport region and as a place of metabolism of diverse drugs and chemicals, which helped the scientific community to understand the mechanisms of transport and metabolism through the placenta, while the development of the ex vivo placental perfusion method gave us a significantly clearer picture of the toxicokinetics of the placenta [133].

Another complementary tool in the hands of scientists to clarify the mechanism of transport and metabolism of drugs and chemicals through the placenta has been modern computer-aided technologies. These technologies are also used to predict some possible preclinical toxicological results and side effects, as all of the above are vital in the drug discovery process. In particular, computer-aided toxicology (Figure 5) is commonly used in drug development programs with the aim of producing safe drugs. In this aspect, several software packages based on QSAR methodology have been used to analyze the chemical structure of compounds and correlate this structure with the potential activity or toxicity of these compounds [134,135].

Models on placental clearance index prediction for diverse drugs and chemicals have been proposed using the ex vivo diffusion technique, QSAR-based studies, and multivariate data analysis (MVDA) [102,103,104,105,106,107,136]. Reliable QSAR models have been created based on fundamental characteristics of molecules, such as whether they are polar or hydrophobic or have halogen atoms as substituents [136,137]. However, those models used in reproductive toxicology based on QSAR methodology to estimate the extent of transfer of xenobiotics may not always be so reliable since they accept that the transport of drugs and chemicals through the placental barrier occurs only by passive diffusion and not by active transport [102,103,104,105,106,107,136,137]. Another disadvantage of QSAR models is that the derived estimation may ultimately refer to the metabolite rather than the parent compound as it does not take into account the metabolism that the xenobiotic may be taken in by the placenta. Modeling tools such as MetaDrug, MetaSite, and TIMES can help in obtaining more reliable results [136,137,138].

In addition to QSAR models, there is one more computer-aided tool that could help scientists to understand and elucidate the placental transport/metabolism mechanisms of drugs/chemicals. This physiology-based tool is pharmacokinetic (PBPK) modeling. Studies report that the PBPK model can be applied to predict the course of all parent substances, metabolites, and some biomarkers in the studied organism, incorporating mechanistic data related with the absorption, distribution, metabolism, and excretion (ADME) properties [134,135,136,137]. The transport and elimination process of drugs and their metabolites can be influenced by some physiological factors such as blood flow rate, flow pattern, and intestinal transit but also by biochemical factors such as protein binding, transporters, and enzymes. PBPK modeling has successfully been used to describe these potential factors [138,139].

In the PBPK model, it is considered that the absorption of the drugs or chemicals can be achieved either by passive diffusion or with the help of transporters. The further path of the drug or its metabolites can be either their intracellular metabolism or their excretion with the help of transporters [138,139]. In addition, with the PBPK model, it is possible to estimate the effect of crucial mechanisms and factors regarding tissue dose, thus facilitating the integration of data, from in vitro and in silico methods, so that the best tissue dosimetry for the animals can be predicted. Nevertheless, the application of the PBPK model in terms of the study of the transport and metabolism mechanisms of drugs/chemicals across the placenta is currently under consideration as its application requires complete data from many processes of the organism concerning both physiological and pathophysiological conditions.

Moreover, PBPK models can incorporate physiological, preclinical, and clinical data into the model to predict drug exposure during pregnancy [140,141]. These modeling strategies offer a promising approach to identify the PK changes of drugs or chemicals during the period of pregnancy to guide dose optimization when there is lack of clinical data [122,124]. Apart from maternal–fetal PBPK models, animal studies also have the potential to estimate fetal drug or chemical exposure by allometric scaling [140,141]. However, whether such scaling will be successful remains yet to be determined [140,141]. All of the above evidence suggest that in vivo, in vitro, and ex vivo studies along with these two computer-aided tools, PBPK and QSAR, could be used by researchers to more accurately assess human fetal exposure. A more recent study applied Monte Carlo optimization method-based QSAR modeling, which could be considered as an effective complementary tool for the high-throughput screening of the placental permeability of drugs [142]. However, we should emphasize that computer-aided tools can exclusively be applied as a supplementary method that can offer us only predicted data concerning placental transfer and metabolism of the xenobiotic compounds, and especially for those compounds for which we have no experimental data yet.

In the last few years, the future development of drug conjugates or other macromolecular medicines that can safely be used during gestation has been recommended [134,135,136,137,138,139,140,141,142,143]. This recommendation was initially derived from the observation that the molecular size affected transplacental transport in an in vitro model, BeWo b30 monolayers, as well as in pregnant mice, with larger polymers presenting poor permeability. In addition to molecular size, polymer chemistry altered behavior, with polyethylene glycol (PEG) molecules permeating the placental barrier to a greater extent than dextrans of equivalent molecular weight [140,141]. Moreover, despite the increasing awareness of the ubiquity of microplastics (MPs) in our environments, little is known so far about their risk of developmental toxicity. Even less is known about the environmental distribution and associated toxicity of nanoplastics (NPs). Notably, a systematic review, including 11 research articles covering in vitro, in vivo, and ex vivo models and observational clinical studies concerning the placental translocation of MPs and NPs, has been performed. The above study confirmed that the placental translocation of MPs and NPs may closely be depended on their physicochemical properties such as molecular size, charge, and chemical modification as well as protein corona formation [140,141]. This study has revealed that there is emerging evidence of placental and fetal toxicity due to plastic particles based on animal and in vitro studies [140,141]. Indicatively, field research has confirmed the presence of diverse types of fibrous, spherical, and fragmentary MPs in fish gastrointestinal tracts and gills, and specifically in their muscle and liver [143]. Notably, a recent study provided substantial evidence for the negative effects of NPs on human placental cells, highlighting the need to perform risk assessment studies of the impact of NPs on female reproduction and fetal development [144]. However, more studies are recommended to confirm and quantify the existence of MPs and NPs in human placentas, and especially the potential translocation of different plastic particle types and heterogenous mixtures across the placenta, their exposure in different periods of pregnancy, and the possible associations with adverse birth and other fetus developmental outcomes. Alarmingly enough, the application of nanoparticles in consumer products and nanomedicines has increased dramatically in the last decade, while serious concerns for the nano-safety of susceptible populations are growing [145]. Maternal exposure to NPs during pregnancy leads to adverse gestational parameters, neurotoxicity, reproductive toxicity, immunotoxicity, and respiratory toxicity in offspring [145,146]. Specifically, oxidative stress and inflammation, DNA damage, apoptosis, and autophagy have been considered as the main mechanisms underlying NP-induced fetotoxicity [147,148]. However, the detailed mechanisms underlying embryo–fetal toxicity need further investigation due to the complexity of the pharmacokinetics of NPs and interactions with physiological systems, particularly during pregnancy [147,148].

## 5. Conclusions

The present review study constitutes a holistic overview of the assessment of drug and chemical transport and metabolism across the human placental barrier. This is a significant and contemporary issue, as during pregnancy, several drugs and chemicals with specific characteristics, such as low polarity and molecular weight, can be transferred to the fetus through the placental barrier by passive diffusion mechanisms, which can be toxic to the fetus. Furthermore, if the drugs or chemicals are similar to endogenous substances, they can also use active placental transporters for their transport, which may considerably affect the exposure of the fetus to them, but also the treatment of a sick fetus. In addition, the important role of enzymes that can metabolize xenobiotics and produce metabolites that may also be toxic to the fetus should be noted. Thus, it is of great importance to critically summarize and scrutinize the currently available experimental methods and techniques that could estimate the potential transport and metabolism of drugs and chemicals such as environmental pollutants, pesticides, and other synthetic compounds. In this aspect, the present holistic overview has highlighted the strengths and limitations of the currently available experimental methods and techniques for the placental transport and metabolism of xenobiotic compounds.

However, based on the present evidence, the molecular processes and mechanisms governing placental transport and metabolism are very complex. Moreover, there is a continuous discovery of novel drugs for which placental transport and metabolism mechanisms should be determined as soon as possible. It is therefore very important for the scientific community to find novel and more advanced methods that will provide an accurate estimation of the mechanisms of transport and metabolism of drugs and chemicals during pregnancy across the human placental barrier so that no risk to the fetus arises. For this reason, several in vivo and in vitro studies have been carried out that have greatly helped us to understand and clarify these mechanisms. In addition, the ex vivo method of placental perfusion provides a much clearer picture of placental toxicokinetics. In silico, computer-aided methods, such as QSAR and PBPK, could also be considered as a complementary tool for the estimation of the fetotoxicity of diverse drugs and chemicals when the above methods are not easily performed. Alarmingly enough, there is a strong demand to assess the potential translocation of different plastic particle types and heterogenous mixtures across the human placenta at different periods of pregnancy, as well as the possible associations with adverse birth and other developmental adverse effects.

## Figures and Tables

**Figure 1 diseases-12-00114-f001:**
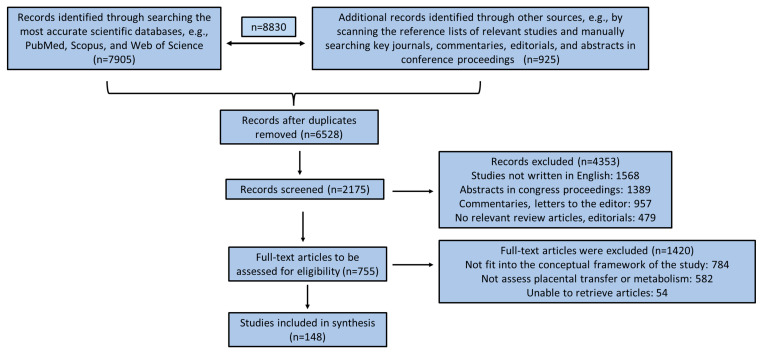
Flow chart diagram for study enrollment based on PRISMA guidelines.

**Figure 2 diseases-12-00114-f002:**
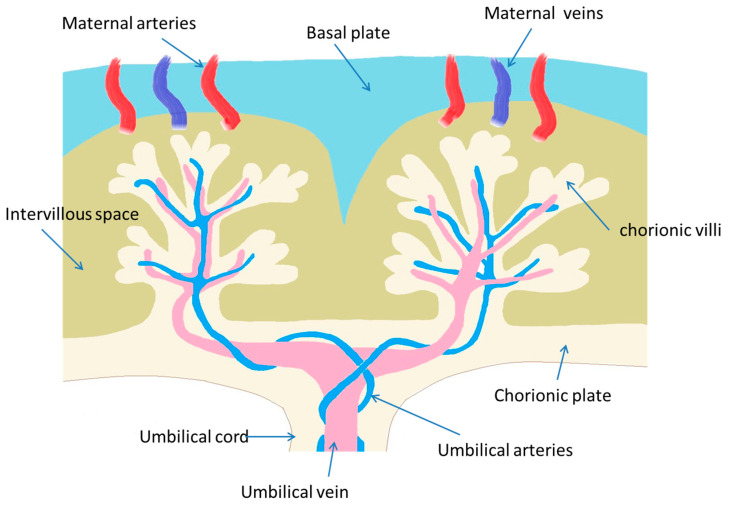
A representative illustration of placenta morphology.

**Figure 3 diseases-12-00114-f003:**
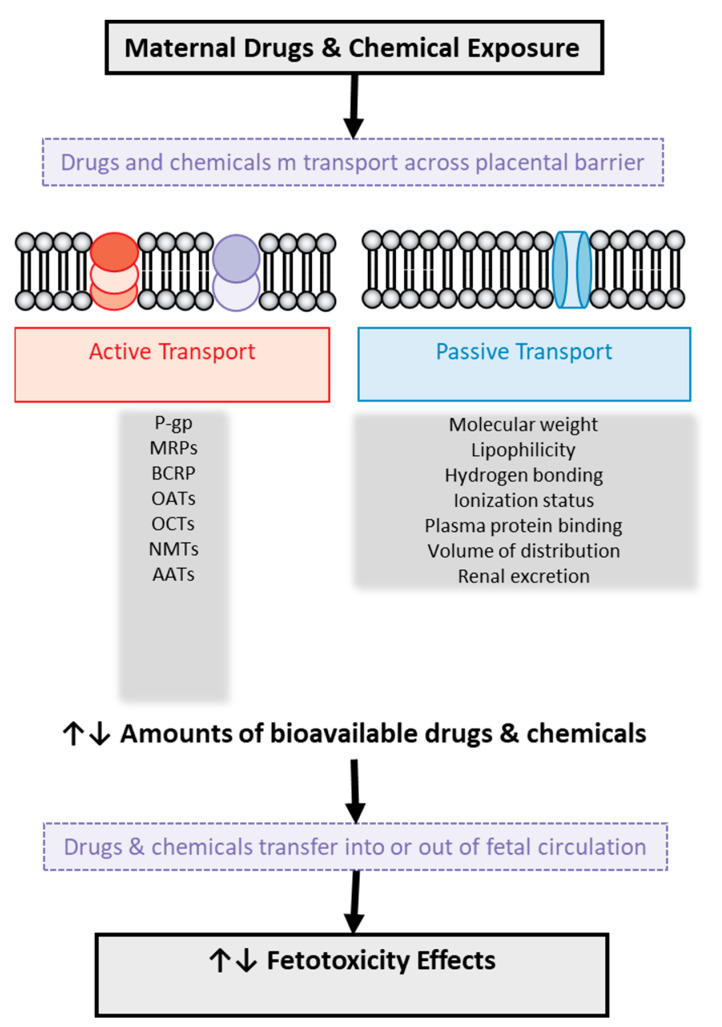
Illustration of drug and chemical transport across the human placental barrier and factors involved in it with potential toxic effects on the fetus. P-gp: P-glycoprotein, MRPs: multidrug resistance-associated protein, BCRP: breast cancer-resistant protein, OATs: organic anion transporters, OCTs: organic cation transporters, NMTs: neuronal monoamine transporters, AATs: amino acid transporters.

**Figure 4 diseases-12-00114-f004:**
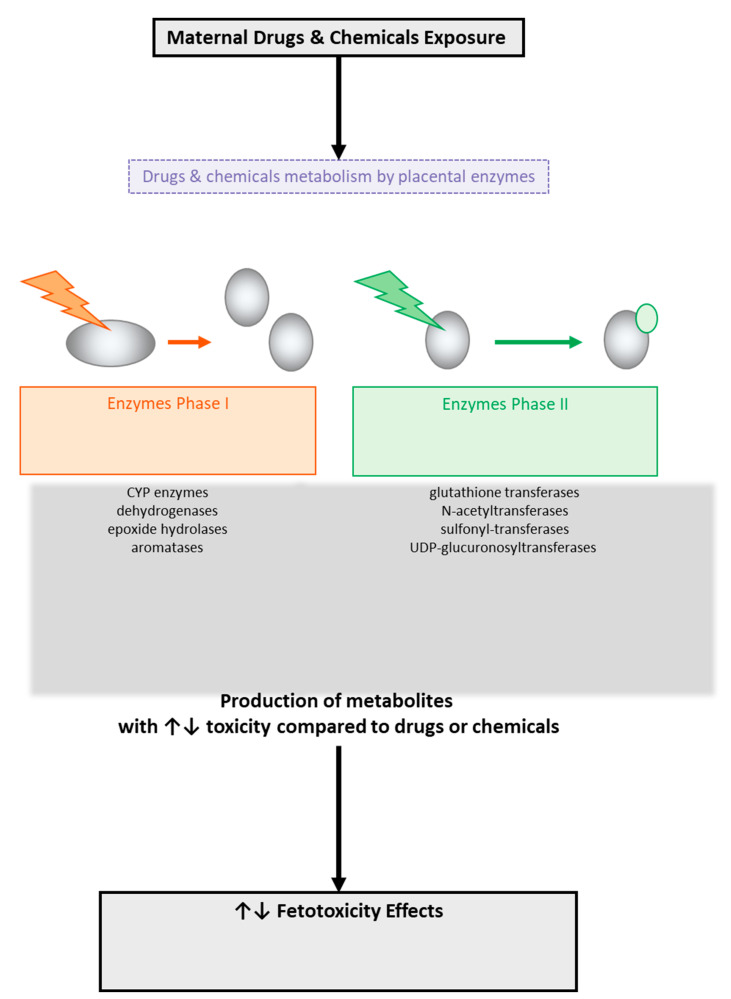
Metabolism of drugs and chemicals across the human placental barrier by placental enzyme-producing metabolites with potential toxic effects on the fetus.

**Figure 5 diseases-12-00114-f005:**
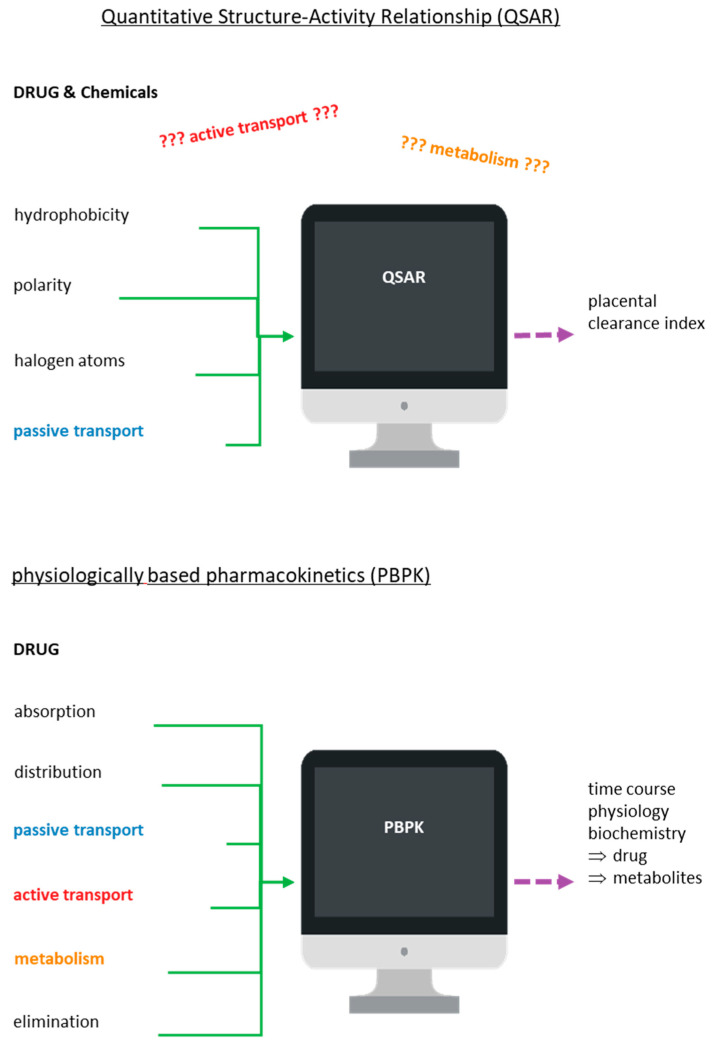
Schematic illustration of the quantitative structure–activity (or toxicity) relationship (QSAR/T) model and the physiologically based pharmacokinetics (PBPK) model.

**Table 1 diseases-12-00114-t001:** Advantages and disadvantages of placental transfer assessment methods.

Placental Transfer Assessment Methods	Advantages	Disadvantages
In vitro studies	Constitute an effective approach simulating in vivo animal models. Reduced costs compared with in vivo animal studies. More suitable to investigate prenatal cell toxicity and the risk of chromosomal abnormalities.	Exhibit low viability.
Ex vivo perfusion technique	The most dependable, accurate, informative, and metabolically stable approach for assessing placental transfer and metabolism. Investigates placental transfer, directly eliminating the need for extrapolation from animal models. Preserves the placental structure entirely, enabling comprehensive insights into various transporters associated with active transport and tissue binding.	Time-consuming method with a relatively low success rate. Not suitable for longer perfusion periods (>12 h) required for studying the transmission of infectious diseases. Utilizes full-term placentas and does not accurately represent the first-trimester placenta. Necessitating a consistent supply of placentas. Not appropriate for large-scale studies at a population level.
Animal studies	Provide quite sufficient data about the placental transfer and the potential toxic effects of xenobiotics on the fetus. Help to design novel targeted therapies.	More costly and raise serious ethical issues. There is serious concern about the relevance of experimental data from animal placental studies to human conditions. Distinct structural and functional differences between the human placenta and those of other mammals.
Human studies	The assessment of the feto-maternal concentration ratio is a straightforward and ethically acceptable technique.	Ethical considerations.

**Table 2 diseases-12-00114-t002:** Transporters that can facilitate or impede the passage of drugs or chemicals from the mother to the fetus.

Transporters	Transport Mechanism	Drug Transporter
P-glycoprotein (P-gp)	Active transport	Transports hydrophobic amphipathics, such as anticancer, immunosuppressive, and cardioprotective drugs, but also steroids and lipid-lowering agents. Blocks calcium channel blockers used to treat gestational hypertension.
Multidrug resistance-associated proteins (MRPs)	Active transport	MRP1 prevents the entry of glutathione/glucuronide metabolites. MRP5 transports cyclic nucleotides (especially cGMP).
Breast cancer-resistant protein (BCRP)	Active transport	Provides resistance against anthracycline antitumor drugs and mitoxantrone. Nitrofurantoin, cimetidine, and glyburide are substrates of BCRP. Pregnancy-related steroid hormones, growth factors, and cytokines regulate BCRP expression. Protective role in removing cytotoxic drugs or chemicals from fetus.
Organic anion transporters (OATs)	Active transport	Transfer organic anions like steroid sulfates derived from cigarette smoking. OATP2B1 contributes to transepithelial transport of steroid sulfates. Transporters of drugs such as anticancer drugs (e.g., methotrexate), non-steroidal anti-inflammatory drugs, and antibiotics.
Organic cation transporters (OCTs)	Active transport	Transporters of carnitine, procaineamide and ofloxacin, clonidine and verapamil. OCTN3 removes catecholamines from the fetus.
Neuronal monoamine transporters (NMTs)	Active transport	Serotonin, dopamine, norepinephrine, and epinephrine constitute the most important physiological substrates of NMTs. Antidepressants and cocaine act as inhibitors of NMTs. Amphetamines are substrates that can be transported, inhibiting NMTs’ capacity to transfer endogenous substrates.
Amino acid transporters (AATs)	Active transport	Facilitate the transportation of xenobiotics. Transport certain pharmaceuticals that have similar chemical structure with amino acids, such as gabapentin.
Equilibrative nucleoside transporters (ENTs)	Active transport	Transport antiviral nucleoside analog drugs, such as didanosine, zalcitabine, and zidovudine, as well as certain anticancer drugs like cytarabine and gemcitabine.

**Table 3 diseases-12-00114-t003:** Enzymes for metabolism of specific drugs.

Chemical/Drug	Enzymes	Organ/Tissue
Theophylline and caffeine	Suppress the microsomal enzymes that metabolize drugs.	Liver
Tobacco	P450 monooxygenase CYP1A1. UDP-glucuronosyltransferase (UGT) activity is stimulated (by both tobacco and alcohol).	Placenta
Glucocorticoids	Suppress the activity of certain placental xenobiotic-metabolizing enzymes.	Placenta
Azidothymidine	Increases CYP1A1, CYP reductase, β-glucuronidase, and GST activities.	Placenta
Phenobarbital	Induces metabolic enzymes in the liver but has no effect on placental enzyme activity.	Liver
Environmental chemical pollutants	CYP1A1 and placental glutathione.	Placenta

## Data Availability

The data of the present study are available upon request to the corresponding author.

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
