# Peer review of "Fetus Exposure to Drugs and Chemicals: A Holistic Overview on the Assessment of Their Transport and Metabolism across the Human Placental Barrier"

_diseases, 2024, doi:10.3390/diseases12060114_

Round 1

Reviewer 1 Report

Comments and Suggestions for Authors

The manuscript proposed by Kotta-Loizou and co-workers, entitled „Fetus exposure to drugs and chemicals: Potential molecular processes and mechanisms on their transport and metabolism across human placenta” (diseases-2953362) is a review presenting potential molecular processes and mechanisms that govern maternal drug and chemical exposure during pregnancy, potentially leading to fetotoxicity effects.

The manuscript is interesting, but it needs to be modified before it can be published.

Kindly find below my principal remarks. 

- The abstract does not adequately present the novelty of this study.,

- abstract does not present the applicability and the importance of the presented work,

- the data presenting placental transfer as a method for estimating fetal toxicity can be summarized as a table

- in my opinion, the paragraph presenting the analytical methods commonly used in the analysis of xenobiotics and their metabolites in human placenta (or in general, the methods used in such analysis), should be presented.  One such technique is mass spectrometry. Its applicability should be described.

- In my opinion, the work does not focus on the topic presented in the title, it touches on many topics but does not reflect the essence of the topic.

- In my opinion, the keywords used in the study should also include, i.e. placental analysis, placental metabolomics, to better present the idea of this work. Therefore, I feel that the manuscript is incomplete.

- The text is well-organized; however, it lacks significant paragraphs that ought to be included.

- The language is generally correct.

 The work is properly prepared from a technical perspective.

Comments on the Quality of English Language

The language is generally correct. Moderate English editing is needed. 

Author Response

Response to Reviewer 1

Comments and Suggestions for Authors

The manuscript proposed by Kotta-Loizou and co-workers, entitled „Fetus exposure to drugs and chemicals: Potential molecular processes and mechanisms on their transport and metabolism across human placenta” (diseases-2953362) is a review presenting potential molecular processes and mechanisms that govern maternal drug and chemical exposure during pregnancy, potentially leading to fetotoxicity effects.

The manuscript is interesting, but it needs to be modified before it can be published.

Kindly find below my principal remarks. 

- The abstract does not adequately present the novelty of this study.

Response: We have now revised the abstract to present the novelty of the study more effectively.

- abstract does not present the applicability and the importance of the presented work.

Response: We have now revised the abstract to present the applicability and the importance of the presented work more effectively.

- the data presenting placental transfer as a method for estimating fetal toxicity can be summarized as a table

Response: Thank you very much for your useful suggestion. We have now a relevant representative table for drugs or chemicals mechanisms of transportation

- in my opinion, the paragraph presenting the analytical methods commonly used in the analysis of xenobiotics and their metabolites in human placenta (or in general, the methods used in such analysis), should be presented.  One such technique is mass spectrometry. Its applicability should be described.

Response: Thank you very much for your useful suggestion. We have now added a separate section entitled as “3.2.5 Analytical techniques for evaluating xenobiotic maternal and fetal exposure”, in which we have presented the analytical methods commonly used in the analysis of xenobiotics and their metabolites in human placenta. This new section was based by the addition of several relevant references.

- In my opinion, the work does not focus on the topic presented in the title, it touches on many topics but does not reflect the essence of the topic.

Response: According to the useful reviewer suggestion, we have now revised the title in order to present more effectively the topic of our manuscript.

- In my opinion, the keywords used in the study should also include, i.e. placental analysis, placental metabolomics, to better present the idea of this work. Therefore, I feel that the manuscript is incomplete.

Response: We have now revised the keywords by including the terms placental analysis, and placental metabolomics.

- The text is well-organized; however, it lacks significant paragraphs that ought to be included.

Response: We have now added certain additional paragraphs such as those concerning the analytical techniques used for evaluating xenobiotic maternal and fetal exposure. Several other statements have also been added throughout the manuscript, providing more specifications and explanations concerning the topic of our review article.

- The language is generally correct.

Response: Thank you very much for your kind words.

 The work is properly prepared from a technical perspective.

Response: Thank you very much for your kind words.

Comments on the Quality of English Language

The language is generally correct. Moderate English editing is needed.

Response: We have checked throughout the manuscript to revise typos and language errors, as well as syntax/grammar errors.

Reviewer 2 Report

Comments and Suggestions for Authors

Prescription medication is inevitable during pregnancy. However, the list of drugs that can be taken during pregnancy is severely limited due to the sensitivity of the organ. A thorough discussion of pharmacological and chemical exposure to the fetus, as well as its transportation and metabolism through the human placenta, is necessary. Kotta-Loizou et al., have submitted a comprehensive review entitled “Fetus exposure to drugs and chemicals: Potential molecular processes and mechanisms on their transport and metabolism across human placenta.”

The authors did a good job in comprehending this review by considering the vast literature, encompassing in vitro, in vivo and human studies. The manuscript is contentious and flawless.

3.1 Placenta anatomy and function. This is an instrumental introduction and needs to be increased its elegance by constructing a representative figure.

A list of important drugs and chemicals should be tabulated with their mechanism of transportation and metabolism.

Author Response

Response to Reviewer 2

Comments and Suggestions for Authors

Prescription medication is inevitable during pregnancy. However, the list of drugs that can be taken during pregnancy is severely limited due to the sensitivity of the organ. A thorough discussion of pharmacological and chemical exposure to the fetus, as well as its transportation and metabolism through the human placenta, is necessary. Kotta-Loizou et al., have submitted a comprehensive review entitled “Fetus exposure to drugs and chemicals: Potential molecular processes and mechanisms on their transport and metabolism across human placenta.”

Response: We have now added some statements in the introduction section to emphasize the fact that prescription medication is inevitable during pregnancy, while the list of drugs that can be taken during pregnancy is severely limited due to the sensitivity of the organ. We have also added a statement in the introduction section to highlight that a thorough discussion of pharmacological and chemical exposure to the fetus, as well as its transportation and metabolism through the human placenta, is necessary.

The authors did a good job in comprehending this review by considering the vast literature, encompassing in vitro, in vivo and human studies. The manuscript is contentious and flawless.

Response: Thank you very much for your kind words.

3.1 Placenta anatomy and function. This is an instrumental introduction and needs to be increased its elegance by constructing a representative figure.

Response: Thank you for your useful suggestion. We have now added a representative figure for placenta morphology.

A list of important drugs and chemicals should be tabulated with their mechanism of transportation and metabolism.

Response: Thank you for your useful suggestion. We have now added two representative tables for drugs and chemicals mechanisms of transportation and metabolism.

Reviewer 3 Report

Comments and Suggestions for Authors

In this comprehensive, literature review article, the authors summarized and scrutinized the potential molecular processes and mechanisms governing maternal xenobiotics’ exposure during pregnancy, which subsequently lead to deleterious effects to the fetus.

Comments

This is an interesting review article. The manuscript is well-written. The reviewer has only some minor concerns as follows:

1.     For “3.2 Evaluating placental transfer as a method for estimating fetal toxicity” section, a summary table comparing different approaches and their pros and cons can increase the reader's understanding of this topic.

2.     In Figures 1-3, the font within the figure could be larger to make it easier to read.

3.     In Figure 1, the full names of the abbreviations can be listed in the footnote.

Author Response

Response to Reviewer 3

Comments and Suggestions for Authors

In this comprehensive, literature review article, the authors summarized and scrutinized the potential molecular processes and mechanisms governing maternal xenobiotics’ exposure during pregnancy, which subsequently lead to deleterious effects to the fetus.

Comments

This is an interesting review article. The manuscript is well-written. The reviewer has only some minor concerns as follows:

Response: Thank you very much for your kind words.

  1. For “3.2 Evaluating placental transfer as a method for estimating fetal toxicity” section, a summary table comparing different approaches and their pros and cons can increase the reader's understanding of this topic.

Response: Thank you for your suggestion. We have now added a table with the advantages and disadvantages for the placental transfer methods for estimating fetal toxicity.

  1. In Figures 1-3, the font within the figure could be larger to make it easier to read.

Response: We have now increased the size of Figures 1-3 (Figures 2-4 in revised manuscript).

  1. In Figure 1 the full names of the abbreviations can be listed in the footnote.

Response: We have now included the full names of the abbreviations of Figure 1 (Figure 2 in revised manuscript), in its footnote.

Reviewer 4 Report

Comments and Suggestions for Authors

Thank you very much for allowing me to review the article titled “Fetus exposure to drugs and chemicals: Potential molecular processes and mechanisms on their transport and metabolism across human placenta” (diseases-2953362).

In the abstract, the objective of the study should be clearly and precisely stated. In the methodology section, it should indicate the review period and the number of articles used, at a minimum. The methodology should be a single section, not two. It is difficult to evaluate the conclusion, as it is very nonspecific, since the objective does not appear.

The introduction addresses the importance of chemicals that can cross the placental barrier and affect both the mother and the fetus. The objective stated is "a comprehensive review aims to critically summarize and scrutinize the potential molecular processes and mechanisms governing maternal xenobiotics’ exposure during pregnancy, which subsequently lead to deleterious effects on the fetus." Therefore, the review is focused on chemicals such as pollutants, drugs, pesticides, and other synthetic compounds. This is indeed a very broad objective that would require a more focused introduction on these topics.

In the methodology section, it is indicated that two databases have been used, but the period reviewed is not specified. I do not understand why computer-aided studies are being used, as there are many studies in these systems that are not scientific. The articles found and ultimately used are not specified; generally, the PRISMA system is used to present this information, and I suggest the authors use it.

The results section begins with the Structure of the placenta, which I do not believe should be part of the results. The same applies to section two. The rest of the results are structured and contain relevant information.

Regarding the discussion, the lack of bibliographic references is striking, as they are very important in this section. Furthermore, given the complexity of the topic, it seems inadequate.

The conclusion is the contribution of this review. The authors present it as a summary, but it lacks depth given the complexity of the subject matter.

Author Response

Response to Reviewer 4

Comments and Suggestions for Authors

Thank you very much for allowing me to review the article titled “Fetus exposure to drugs and chemicals: Potential molecular processes and mechanisms on their transport and metabolism across human placenta” (diseases-2953362).

In the abstract, the objective of the study should be clearly and precisely stated.

Response: We have now revised the abstract in several point, reporting also more clearly and precisely the objective of our study. We have also performed some more specification to improve the quality of the abstract.

 In the methodology section, it should indicate the review period and the number of articles used, at a minimum. The methodology should be a single section, not two.

Response: We have now added a statement in the methods section, reporting that no time limitation was applied for the final selection of the studies included in the present review. We have also added a statement that after applying all the above exclusion criteria in conjunction with the articles retrieved from the full text of the initial selected articles, a number of 148 published articles were included in the present review study.  We have also added a statement to emphasize that the first search in the three scientific databases revealed 8830 published articles until 5th February 2024.

It is difficult to evaluate the conclusion, as it is very nonspecific, since the objective does not appear.

Response: We have now added several statements in the conclusion section, emphasizing the objective and the usefulness of our review study, and providing more specifications, and explanations  concerning the topic of our review article.

The introduction addresses the importance of chemicals that can cross the placental barrier and affect both the mother and the fetus. The objective stated is "a comprehensive review aims to critically summarize and scrutinize the potential molecular processes and mechanisms governing maternal xenobiotics’ exposure during pregnancy, which subsequently lead to deleterious effects on the fetus." Therefore, the review is focused on chemicals such as pollutants, drugs, pesticides, and other synthetic compounds. This is indeed a very broad objective that would require a more focused introduction on these topics.

Response: We have now added some statements in the introduction section to emphasize the fact that prescription medication is inevitable during pregnancy, while the list of drugs that can be taken during pregnancy is severely limited due to the sensitivity of the placenta. We have also added a statement in the introduction section to highlight that a thorough discussion of pharmacological and chemical exposure to the fetus, as well as its transportation and metabolism through the human placenta, is necessary.

In the methodology section, it is indicated that two databases have been used, but the period reviewed is not specified. I do not understand why computer-aided studies are being used, as there are many studies in these systems that are not scientific. The articles found and ultimately used are not specified; generally, the PRISMA system is used to present this information, and I suggest the authors use it.

Response: In the methodology section, we have reported that we used three scientific databases, e.g. PubMed, Scopus, and Web of Science. These are the most accurate and reliable scientific databases worldwide. We have also added a statement in the methods section, reporting that no time limitation was applied for the final selection of the studies included in the present review. We have also added a statement that after applying all the above exclusion criteria in conjunction with the articles retrieved from the full text of the initial selected articles, a number of 148 published articles were included in the present review study.  We have also added a statement to emphasize that the first search in the three scientific databases revealed 8830 published articles until 5th February 2024. We did not include the computer-aided studies in the main text of our manuscript but only in the discussion section. We added that the computer-aided tools could exclusively be applied as a supplementary method, which can offer us only predict-ed data concerning placental transfer and metabolism of the xenobiotic compounds, and especially for those compounds for which we have not experimental data yet.

The results section begins with the Structure of the placenta, which I do not believe should be part of the results. The same applies to section two. The rest of the results are structured and contain relevant information.

Response: In an effort to include all the comments and suggestions from the four reviewers of our manuscript and based on the suggestions of the other three reviewers, we have changed the subtitle of the subsection 3.1 of the results section in order to be more representative with its content and we have kept this subsection at the results section.

Regarding the discussion, the lack of bibliographic references is striking, as they are very important in this section. Furthermore, given the complexity of the topic, it seems inadequate.

Response: We have now added several bibliographic references in the discussion section. Moreover, we have now added several statements and their relevant references to enhance the adequacy of the discussion section.

The conclusion is the contribution of this review. The authors present it as a summary, but it lacks depth given the complexity of the subject matter.

Response: We have now added several statements in the conclusion section to emphasize the complexity of the subject matter.

Round 2

Reviewer 1 Report

Comments and Suggestions for Authors

The revised version of the manuscript presented by Kotya-Loizou and co-workers meets my requirements. The document was significantly improved, the Authors presented comments and answers to all of my questions and doubts. 

Author Response

Response to Reviewer 1

Comments and Suggestions for Authors

The revised version of the manuscript presented by Kotya-Loizou and co-workers meets my requirements. The document was significantly improved, the Authors presented comments and answers to all of my questions and doubts.

Response: Thank you very much for your kind words and your useful comments and suggestions, which help us to improve our manuscript.

Reviewer 4 Report

Comments and Suggestions for Authors

I have thoroughly reviewed the latest version of the manuscript “Fetus exposure to drugs and chemicals: Potential molecular processes and mechanisms on their transport and metabolism across human placenta” (diseases-2953362).as well as the authors' response letter to the previous comments. The authors have made significant improvements to enhance the clarity and comprehensibility of their work. However, there are still some areas that need further refinement, as I mentioned in my earlier comments.

In the abstract, it is essential to specify the time period covered in the literature review and the number of articles that meet the inclusion criteria. Even though all relevant articles have been included, it should be explicitly stated that the review extends up to 2024. Additionally, in the methodology section, the phrase "A comprehensive and in-depth research" should be clarified to indicate that it refers to a review.

Including the well-known structure of the placenta, which is already extensively covered in standard textbooks, does not contribute new information and therefore seems unnecessary.

In the methodology section, the selection process for the articles should be presented using a selection diagram. I recommend employing the PRISMA guidelines for this purpose.

Author Response

Response to Reviewer 4

Comments and Suggestions for Authors

I have thoroughly reviewed the latest version of the manuscript “Fetus exposure to drugs and chemicals: Potential molecular processes and mechanisms on their transport and metabolism across human placenta” (diseases-2953362) as well as the authors' response letter to the previous comments. The authors have made significant improvements to enhance the clarity and comprehensibility of their work. However, there are still some areas that need further refinement, as I mentioned in my earlier comments.

In the abstract, it is essential to specify the time period covered in the literature review and the number of articles that meet the inclusion criteria. Even though all relevant articles have been included, it should be explicitly stated that the review extends up to 2024. Additionally, in the methodology section, the phrase "A comprehensive and in-depth research" should be clarified to indicate that it refers to a review.

Response: Thank you for your useful comments. We have now revised the Abstract and the Methods section according to the above reviewer suggestions.

Including the well-known structure of the placenta, which is already extensively covered in standard textbooks, does not contribute new information and therefore seems unnecessary.

Response: At the first version of the manuscript, we did not include this figure with the structure of the placenta. However, another reviewer has been recommended us to add this figure. For this purpose, we included this figure in the revised version of our manuscript.

In the methodology section, the selection process for the articles should be presented using a selection diagram. I recommend employing the PRISMA guidelines for this purpose.

Response: According to the useful reviewer comment, we have now added a flow chart diagram for the selection process of the articles thaw were presented employing the PRISMA guidelines.
